# Implementation Schemes for Electric Bus Fleets at Depots with Optimized Energy Procurements in Virtual Power Plant Operations

**Andreas F. Raab [1,*], Enrico Lauth [2], Kai Strunz [1] and Dietmar Göhlich [2]**

[1]    Department Sustainable Electric Networks and Sources of Energy (SENSE), Technische Universität Berlin, Einsteinufer 11 (EMH-1), D-10587 Berlin, Germany; kai.strunz@tu-berlin.de

[2]    Department Methods for Product Development and Mechatronics (MPM), Technische Universität Berlin, Straße des 17. Juni 135, D-10623 Berlin, Germany; enrico.lauth@tu-berlin.de (E.L.); dietmar.goehlich@tu-berlin.de (D.G.)

*    Correspondence: andreas.raab@tu-berlin.de; Tel.: +49-(0)177-4221-853

**Abstract:** For the purpose of utilizing electric bus fleets in metropolitan areas and with regard to providing active energy management at depots, a profound understanding of the transactions between the market entities involved in the charging process is given. The paper examines sophisticated charging strategies with energy procurements in joint market operation. Here, operation procedures and characteristics of a depot including the physical layout and utilization of appropriate charging infrastructure are investigated. A comprehensive model framework for a virtual power plant (VPP) is formulated and developed to integrate electric bus fleets in the power plant portfolio, enabling the provision of power system services. The proposed methodology is verified in numerical analysis by providing optimized dispatch schedules in day-ahead and intraday market operations.

**Keywords:** energy demand; electric vehicle; modeling, optimization; power management; public transport; smart charging; smart grid; vehicle to grid

---

## 1. Introduction

The rollout of mobility solutions is accompanied by substantial challenges in the transport and energy sector. This includes, for example, the reduction of the total cost of ownership, the provision of sufficient charging infrastructures, the agreement of standards, and the formulation of regulatory requirements [1,2]. For charging processes and the provision of active energy management, a variety of limiting factors has to be taken into account [3], e.g., distance traveled, road topology, driving behavior, prevailing traffic conditions, and ambient temperature. Additionally, the physical layout and utilization of the charging infrastructure need to be considered. Overall, these factors can have a significant influence to enable similar operational deployments as for conventional vehicles with internal combustion engines.

In multiple pilot projects, different technologies for electric buses and charging systems are currently being tested and demonstrated. Taking Europe for example, different bus types are utilized with conductive or inductive charging systems [1,4]. With the focus on the operation of electric bus fleets, comprehensive analysis for the determination of network capacities and appropriate solutions for the power supply are required [5]. Here, several optimization techniques for smart charging strategies can be adopted to lower the overall energy cost and avoid grid congestion and peak loads caused by charging processes [6–9]. Considering the charging load in enhanced energy management and supply solutions, it is necessary using predictive forecast methods to determine the energy demands [10].

Compared to the existing research as mentioned above and further aggregation and scheduling concepts in [11,12], the paper proposes a solution to integrate electric bus fleets in VPP operations. A methodology for the estimation of the energy demand is carried out by analyzing field-recorded data of diesel demands, determining the energy equivalence and forming bus type-specific vehicle models. Furthermore, charging possibilities are identified that correspond to the operation conditions and services processes at a bus depot. As a result, optimal charging schedules are obtained while making use of these additional energy sources for energy market participation and the provision of power system services. This is achieved thanks to novel VPP functions that exploit the potential of optimized redispatch solutions using the storage capacities of the electric bus fleets at a range of spatial and temporal scales.

First, Section 2 introduces the framework condition for the operational planning and operation of electric bus fleets, specifying the functional roles and responsibilities of entities involved in the charging process. The methodology for estimating the required energy demand is introduced. Section 3 identifies the fundamental characteristics of a depot, including services and processes impacting the charging process. Then, the charging strategies and the value of the proposed methodology for optimized energy procurements in VPP operations are substantiated in Section 4 in numerical simulations. Feasible solutions for the provision of systems services through optimized redispatch measures are presented. Finally, Section 5 contains the concluding remarks.

## 2. Framework Conditions and Operational Planning

Addressing the system complexity for possible implementation schemes for electric bus fleets in liberalized energy markets, clear definitions of functional roles and responsibilities are essential. This may include the introduction of an electric vehicle supplier/aggregator (EVS/A). This market entity collates the energy demand of a number of electric vehicles (EVs) and is responsible for the maintenance planning, operating data acquisition, and management. The EVS/A cooperates with the VPP operator to gain access and visibility across the energy market. Therefore, appropriate charging strategies are negotiated and the commercial conditions defined in advance. Figure 1 gives the framework conditions of the proposed methodology.

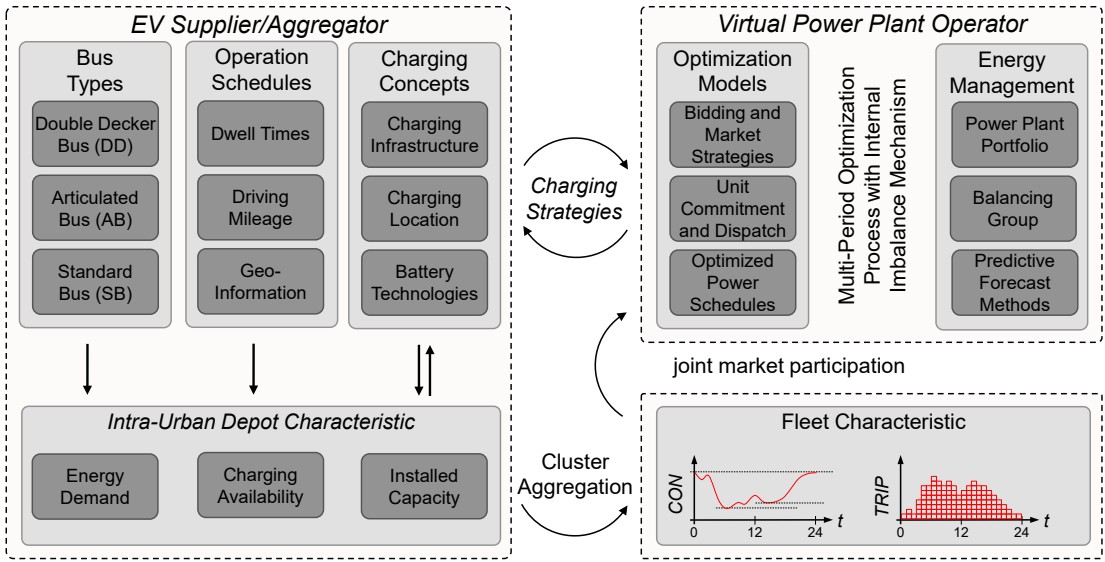

**Figure 1.** Framework conditions for the integration of electric bus fleets in the operation scheme of the EVS/A and VPP operator.

First, the EVS/A determines the energy demand for the operation of an electric bus. The basic characteristics, such as the layout of the depot and spatial availability for charging processes, are considered. Here, internal operational processes and maintenance and service processes are

taken into account as this reduces the availability for charging processes. These framework conditions reveal significant boundary conditions for the optimization problem of the VPP operator, which incorporates the electric bus fleet in the energy management of the power plant portfolio.

*2.1. Modeling Timetable-Based Driving Schedules*

For modeling the driving behavior and determining the required energy demand, operation schedules of specific bus routes of the Berlin metropolitan area are analyzed. In order to capture the main characteristics of the electric bus fleet $H_{\text{fleet}}$, as defined by (1), different vehicle models $H_{\text{mstor}}$ are introduced. For simplification, three different types are considered, namely standard (SB), articular (AB), and double decker buses (DD).

$$\bigsqcup H_{\text{fleet}} = \{(mstor, i) \mid mstor \in H_{\text{mstor}} \ i \in H_{\text{fleet}}\} \tag{1}$$

The operation schedules are composed by several trips for each electric bus $i$, including the departure time at origin $t^{D,O}$, arrival time at destination $t^{A,D}$, and the mileage of driving $m^{OD}$. The event-oriented trips are converted into timetable-based driving profiles by the setting the time increment $\Delta t$ of the operation schedules to 0.25 h. The stepwise approximation allows following typical accounting requirements when participating in liberalized energy markets and the operation of balancing groups. The processing yields the discrete variables $k_{k,i}^{D,O}$ and $k_{k,i}^{A,D}$, defining the time-discrete departure and arrival time, while $m_{k,i}^{OD}$ specifies the mileage of driving. The ***TRIP*** matrix, as generally expressed by (2), contains the resulting discrete variables.

$$\boldsymbol{TRIP} : H_{\text{ts}} \times H_{\text{fleet}} \rightarrow \mathbb{R}^3 \text{ with } \boldsymbol{trip}_{k,i} \mapsto \begin{pmatrix} k_{k,i}^{D,O} \\ k_{k,i}^{A,D} \\ m_{k,i}^{OD} \end{pmatrix} \tag{2}$$

Thereby, the planning horizon for the investigated operation processes is specified by the set of discrete time steps $H_{\text{ts}} = \{k_{\text{ini}}, \ldots, k_{\text{fin}}\}$. The connection matrix ***CON*** gives the binary relation of spatial movement and the temporal availability of the electric buses for charging processes. The logical relation is defined by (3). For example, the element $con_{k,i}$ is equal to one if the $i^{\text{th}}$ bus is connected to the charging infrastructure in the $k^{\text{th}}$ time step and zero otherwise.

$$\{con_{k,i} = 1\} \oplus \{m_{k,i}^{OD} > 0\} = 1 \quad \forall k \in H_{\text{ts}}, \ i \in H_{\text{fleet}} \tag{3}$$

The compact description of the operation schedule is forwarded from the EVS/A to the VPP operator as indicated in Figure 1. Subsequently, the total energy demand of the electric bus fleet is calculated by applying (4) and assigning a specific energy demand for driving $E_{\text{d},k}^{mstor,\text{km}}$.

$$E_{\text{d},k}^{\text{fleet}} = E_{\text{d},k}^{mstor,\text{km}} \cdot \sum m_{k,i}^{OD} \tag{4}$$
$$\forall mstor \in H_{\text{mstor}}, \ \forall k \in H_{\text{ts}}, \ \forall i \in H_{\text{fleet}}$$

The specific energy demand is a representative value for the introduced electric vehicle models $H_{\text{mstor}}$. The assigned model attributes are approximated according to field-recorded data and calculations. Details are provided in the following section.

*2.2. Energy Equivalence and Bus Type-Specific Models*

Field-recorded data of diesel demands, as depicted in Figure 2, are investigated to obtain the energy equivalents for driving and auxiliary components by using the efficiency method. A linear regression model initializes the hypothesis concerning the relationship among the specific energy demand, use of auxiliary devices, and the ambient temperature.

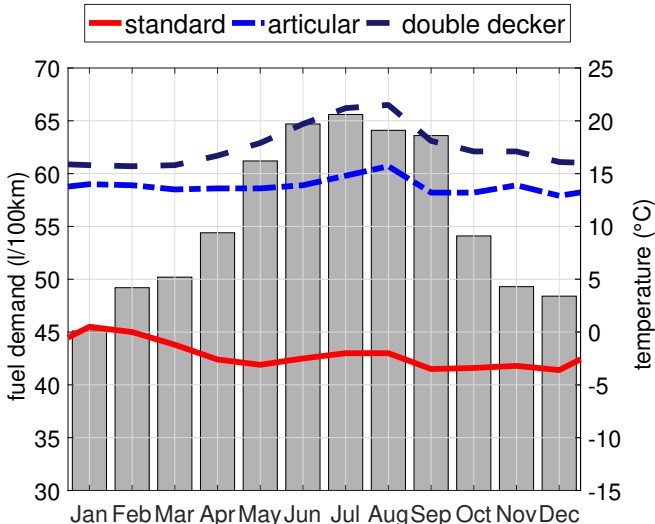

**Figure 2.** Fuel demand of diesel buses (dBus) in accordance with the ambient temperature.

The specific energy demand for driving is assumed to cover the required energy for traction and energy conversion units. First, the average diesel demand $E_{\text{tank}}^{\text{dBus}}$ is calculated by (5) as a function of the diesel demand of an entire fleet $V_{\text{d,diesel}}^{\text{fleet}}$, the lower calorific value of diesel $LCV$, and the total mileage.

$$E_{\text{tank}}^{\text{dBus}} = \frac{V_{\text{d,diesel}}^{\text{fleet}} \cdot LCV}{\sum \sum m_{k,i}^{OD}} \quad \forall k \in H_{\text{ts}}, \forall i \in H_{\text{fleet}} \tag{5}$$

Assuming $\sum \sum m_{k,i}^{OD} = 13 \times 10^6$ km total mileage and $V_{\text{d,diesel}}^{\text{fleet}} = 6.5 \times 10^6$ l diesel demand, the average diesel demand is $E_{\text{tank}}^{\text{dBus}} \approx 4.95$ kWh/km for $LCV = 9.94$ kWh/l. The energy demand for driving $E_{\text{d,drive}}^{\text{eBus}}$ is approximated by (6). The diesel equivalent is multiplied with the tank-motor $\eta_{\text{mot}}^{\text{tank}}$ and motor-drive $\eta_{\text{drive}}^{\text{mot}}$ efficiency. The idling losses $E_{\text{d,idle}}^{\text{dBus}}$ are considered within the approximation.

$$E_{\text{d,drive}}^{\text{eBus}} = E_{\text{d,drive}}^{\text{dBus}} = E_{\text{tank}}^{\text{dBus}} \cdot \eta_{\text{mot}}^{\text{tank}} \cdot \eta_{\text{drive}}^{\text{mot}} - E_{\text{d,idle}}^{\text{dBus}} \tag{6}$$

Finally, the energy demand served by the battery $E_{\text{d,bat}}^{\text{eBus}}$ for the traction process is estimated with (7), where $\eta_{\text{mot}}^{\text{bat}}$ and $\eta_{\text{mot}}^{\text{recu}}$ denote the battery-motor and recuperation-motor efficiency, respectively. The offset values correspond to the energy demand for driving $E_{\text{d,drive}}^{\text{eBus}}$, auxiliary components $E_{\text{d,aux}}^{\text{eBus}}$, and the energy $E_{\text{g,recu}}^{\text{eBus}}$ of the recuperation process.

$$E_{\text{d,bat}}^{\text{eBus,km}} = \frac{E_{\text{d,drive}}^{\text{eBus}}}{\eta_{\text{mot}}^{\text{bat}} \cdot \eta_{\text{drive}}^{\text{mot}}} + E_{\text{d,aux}}^{\text{eBus}} - E_{\text{g,recu}}^{\text{eBus}} \cdot \eta_{\text{mot}}^{\text{bat}} \cdot \eta_{\text{mot}}^{\text{recu}} \tag{7}$$

Possible values for the energy demand of auxiliary components and recuperation process are $E_{\text{d,aux}}^{\text{eBus}} \in \{0.6, 0.9, 1.3\}$ kWh/km [13] and $E_{\text{g,recu}}^{\text{eBus}} = [20, 40]$ kWh/100 km [14]. The chemical, mechanical, and electrical efficiency values are derived from the literature [15–17]. By substituting the total volumetric diesel demand over the total mileage in (5) with the values given in Figure 2 and applying (5)–(7), the energy demand values for driving, auxiliary devices, and recuperation as shown in Figure 3 are obtained. Additional factors, such as the driving behavior, weight loading, rolling resistance, and ambient temperature are considered in different operation scenarios [13].

Figure 3 shows the lower, upper, and total values (red line) of the specific energy demand by using (7) for different scenarios varying the ambient temperature for peak and off-peak hours. In peak hours, for example, there is a higher energy demand as a result of the assumed passenger load and traffic volumes. In heating and cooling mode, the energy demand rises due to the additional operation

of auxiliary devices. The obtained energy equivalents yield the approximates of the model attributes summarized in Table 1.

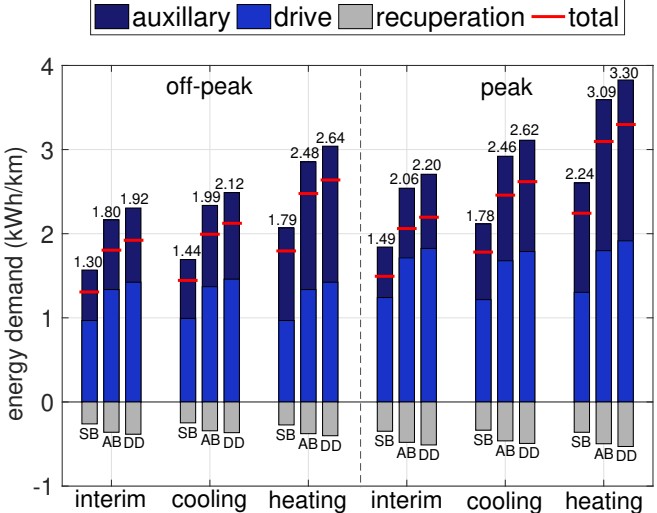

**Figure 3.** Approximated energy demands of electric buses (eBus) summarized for defined operating scenarios.

**Table 1.** Set of unit models $H_{\text{mstor}}$ of electric buses (eBus) specified by the energy capacity, weekly mileage, and specific energy demand.

| Unit Model | Energy Capacity (kWh) | Weekly Mileage of Driving (km) | Specific Energy Demand (kWh/km) | Daily Energy Demand (kWh) |
|---|---|---|---|---|
| SB | 175 | 1045/1147/1469 | 1.30/1.80/2.30 | 245–345 |
| AB | 225 | 1135/1231/1566 | 1.80/2.50/3.10 | 368–508 |
| DD | 250 | 1118/1231/1500 | 1.90/2.60/3.30 | 385–517 |

The model attributes give the descriptive statistic of the analyzed field-recorded data and operation schedules in terms of the weekly mileage and daily energy demand. The assigned energy capacities of the vehicle models with 175, 225, and 250 kWh refer to the usable storage capacity as typically reported in pilot projects [4].

*2.3. Charging Concepts and Strategies*

The following elaborations investigate the depot charging concept combined with opportunity charging. While depot charging reflects charging during the dwell times, opportunity charging takes place on the route, e.g., at terminal stations, by using automated charging systems such as a pantograph or induction system [2]. For the determination of the planned energy demand served by first-base depot charging and second-base opportunity charging at termini, the total energy demand $E_{\text{d},k}^{\text{fleet}}$ for a specific time step $k$ is separated as follows:

$$
\begin{aligned}
E_{\text{d},k}^{\text{fleet}} &= E_{\text{d},k}^{\text{fleet,1st}} + E_{\text{d},k}^{\text{fleet,2nd}} \\
&= \underbrace{\sum_{i \in H_{\text{fleet}}} con_{k,i}^{\text{1st}} \cdot P_{\text{d},k,i}^{\text{1st}} \cdot \eta \cdot \Delta t_{k,i}^{\text{1st}}}_{\text{1st-base charging}} + \underbrace{\sum_{i \in H_{\text{fleet}}} con_{k,i}^{\text{2nd}} \cdot P_{\text{d},k,i}^{\text{2nd}} \cdot \eta \cdot \Delta t_{k,i}^{\text{2nd}}}_{\text{2nd-base charging}} .
\end{aligned}
\tag{8}
$$

The corresponding amount of connected vehicles is given by $CON^{\text{1st}}$ for first-base and $CON^{\text{2nd}}$ for second-base charging. The temporal availability for the charging processes is denoted by $\Delta t_{k,i}^{\text{1nd}}$ and $\Delta t_{k,i}^{\text{2nd}}$, respectively. Different charging infrastructures and modes are represented by taking the charging efficiency $\eta$ into account. However, in the following simulations, the value is assumed to be

equal and set constant. The charging power $P_{\mathrm{d},k,i}^{\mathrm{1st}}$ denotes the contracted charging capacities at the depot within variable charging rates. Due to the limited charging time at termini, the charging capacity $P_{\mathrm{d},k,i}^{\mathrm{2nd}}$ is assumed to be fixed. This is comparable with non-controlled charging processes with maximum charging power. Real operating schedules over one year of a bus fleet are evaluated, pointing out the difference in connectivity and energy demand for first-base and second-base charging. Figure 4a shows the results for a selected week of the entire electric vehicle fleet, while Figure 4b,c give further insight into the number of connected vehicles and corresponding energy demand at these locations.

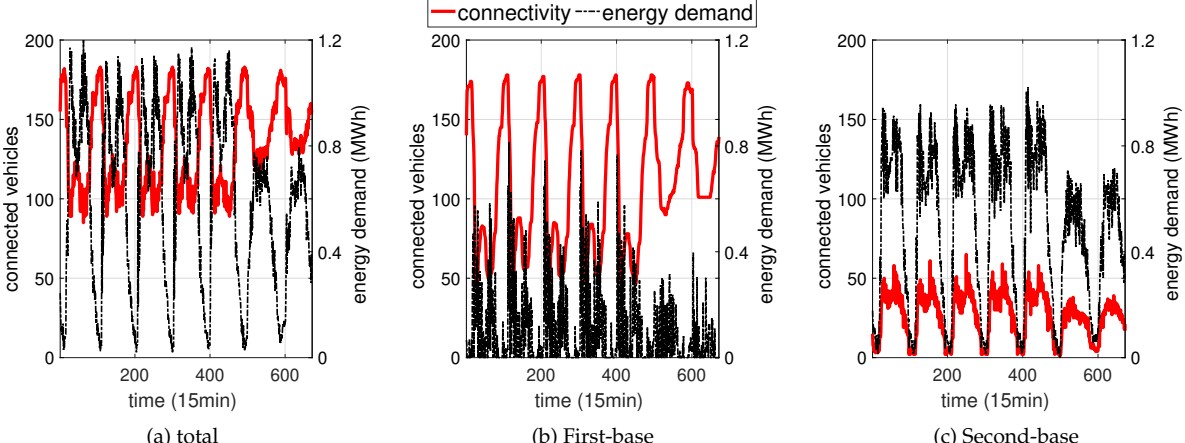

**Figure 4.** Energy demand of an (**a**) arbitrary electric bus fleet separated by energy demand occurring at (**b**) first base charging at the depot and (**c**) second-base charging at termini.

The connectivity of the electric bus fleet is indicated by the red solid lines, while the black dashed lines denote the corresponding energy demands. As can be seen, the amount of available vehicles for first-base charging at the depot is almost three-times higher compared to the available vehicles for the second-base charging at termini. Considering also the dwell times at these charging locations, the elaborations indicate the need to apply different charging concepts and strategies as a function of the expected level of connected vehicles and the total energy demand required.

## 3. Operation Procedures and Depot Characteristics

To gain more insight into the applicable charging concepts and strategies at depots, the operation processes are further investigated and charging possibilities identified. The elaboration yields the input for the case study carried out in Section 4. Figure 5 schematically shows an activity diagram of possible operation processes. Generally, the respective activities are coordinated by the EVS/A by using a depot control center for automatization purposes and for interaction with the VPP operator. The processes are categorized into activities related to the employee responsibilities for the bus driver, service, and workshop staff.

Throughout the processes, three charging possibilities highlighted with dashed rectangles are identified, where generally sufficient time and space are available. First, after entering the depot, the buses are checked for functional capability and then parked in different waiting areas. Afterwards, the service and workshop staff, usually an external service provider, picks up the buses for a required repair, maintenance, or cleaning and refueling. Each bus is required to pass through the daily service process, lasting about 10–15 min [18]. The remaining operation processes are flexible in time and thus also the associated charging possibilities. Since the repair or maintenance path is characterized by unplanned faults of single buses, additional charging possibilities are not explicitly considered here. The second last operation process denotes the parking in a shunting area with sufficient time for charging processes. After the disposition, the buses are ready for the next operation and can leave the depot.

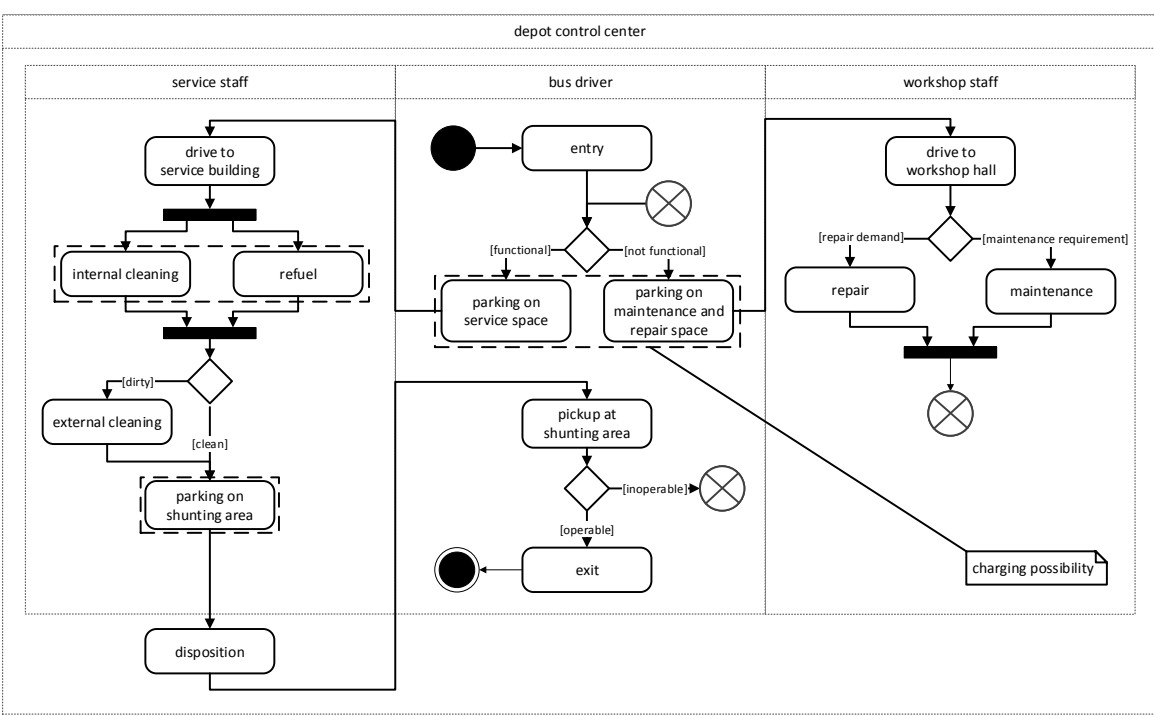

**Figure 5.** Operation processes coordinated by the EVS/A and highlighted charging possibilities at the depot.

### 3.1. Analysis of Operation Processes and Schedules

At the time of possible charging processes, Figure 6 schematically introduces different charging scenarios. Scenario 1 maintains the processes at the depot as introduced by means of Figure 5. Charging is only possible after the service, when the buses are parked in the shunting area. Thus, the time a bus is waiting for service may reduce the possible charging time. Scenario 2 is an extension of the previous scenario. It is assumed that fast charging during the service is possible. This is comparable with conventional refuel processes when considering vehicles with internal combustion engines. Available and tested charging technologies enable conductive fast charging of up to 500 kW, demonstrated in European field sites [4]. Scenario 3 requires an adjustment of the operation processes at the depot. Since usually, buses are cleaned daily only on the inside, this process can take place outside the service hall. Thus, waiting in the service space will be unnecessary, and a bus can be parked directly in the shunting area after arrival and be charged.

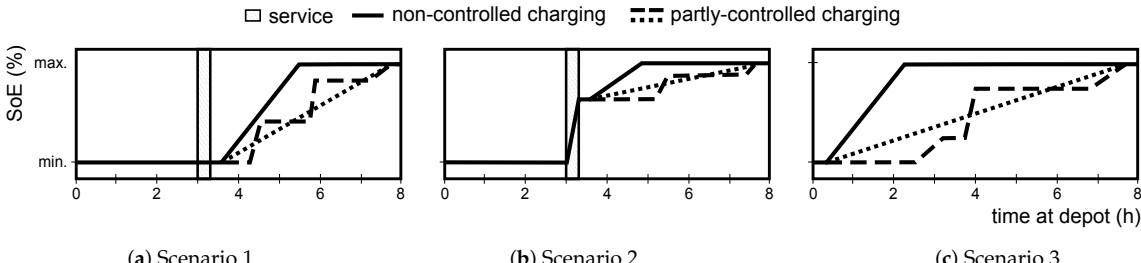

**Figure 6.** Charging process considering the introduced charging scenarios.

Each charging process of the presented scenarios can basically be designed individually. However, for all scenarios, a distinction can be made between non-controlled and partly-controlled charging. Non-controlled charging represents the simplest implementation. Once the bus is connected to the charging infrastructure, the charging process immediately starts with the maximum possible charging power and lasts until the vehicle gets disconnected or the battery is fully charged. Partly-controlled

charging uses the entire dwell time at the depot and uses lower, but also constant charging power. By considering the different charging scenarios, Figure 7 illustrates the differences in the connectivity of the electric bus fleet as a result of the different operation processes.

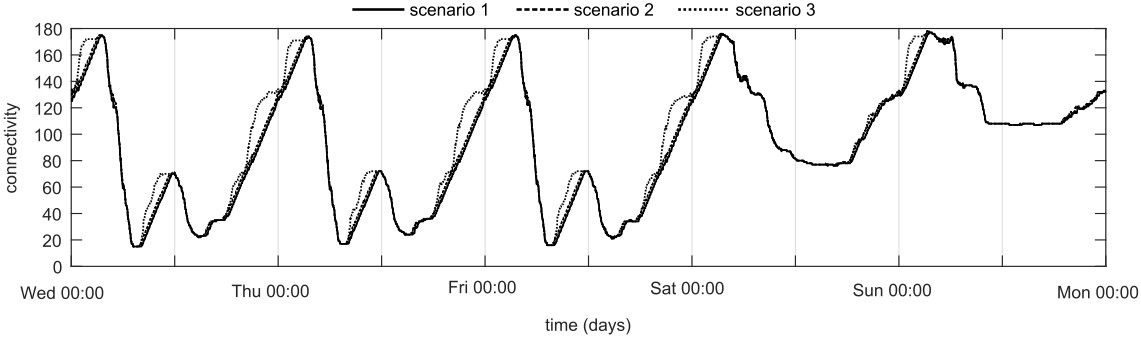

**Figure 7.** Connectivity of buses on selected weekdays and weekends.

During the weekdays, the connectivity profile hardly changes because the operating schedules for Monday–Friday are identical. As can be seen, there are several peaks in the connectivity profile, e.g., after midnight due to the main pause in operation of the fleet and small peaks at noon. The latter is a result of returned buses, which are additionally used for the rush hour in the morning. Compared to the weekdays, the connectivity of the buses rises on the weekend, especially during the day, as the overall utilization of the electric bus fleet is lower. The lowest connectivity can be observed in Scenario 1, since the idle time before the service reduces the possible charging time. The connectivity of Scenario 2 is comparable to Scenario 1, as only the service time is additionally used. The highest connectivity is given in the Scenario 3, because no congestion queue appears during the cumulated return of the buses to the depot. On the weekend, there is a marginal difference. This is a result of fewer buses used during the day and, on the other hand, the extended end time of the operation.

*3.2. Charging Infrastructure and Process*

For the design of the charging infrastructure and the impact on charging schedules, the charging process as introduced with Scenarios 1–3 is further investigated. The scenarios are compared using real operating schedules as utilized in Section 2. In total, a sample set of 193 buses is considered using the unit model attributes specified in Table 1. The service process, as shown in Figure 5, is applied and assumed to be processed by using the first in-first out method. The amount of required charging points yields the maximum connected vehicles according to Figure 7 to serve the energy demand calculated by (8) at the depot.

For the given time period of 24 h, the charging capacity of each charging point is determined. Scenario 1 uses charging points with $P_d^{1st} = 125$ kW each. In Scenario 2, charging points with $P_d^{1st} = 75$ kW and four additional charging points with $P_d^{1st} = 300$ kW are considered. The number of four additional charging points is derived from the maximum number of available service points. The charging power of each charging point in Scenario 3 is $P_d^{1st} = 75$ kW.

Since the potential charging time in Scenario 1 is lower compared to Scenario 3, a higher charging power per charging point is necessary to guarantee the operational processes of the electric bus fleet. By utilizing fast charging in Scenario 2, part of the required energy can be already supplied during the service. Therefore, the charging power of the remaining charging points may be lower, which corresponds to the underlying assumptions of Scenario 3. Figure 8 provides the results achieved by assessing these conditions for the charging processes.

The charging loads of Scenario 1 and Scenario 2 show marginal deviations for the non-controlled charging. The successive charging processes during the service allows reducing the peaks of the charging loads and balancing of the overall charging power, e.g., on workdays in the evening and night hours. As can be seen in all three scenarios, partly-controlled charging does not necessarily

reduce the charging loads and overall peak loads. The charging processes and energy supply are mainly moved to the night hours. Only on weekends, partly-controlled charging leads to a more even distribution of the charging power. However, to provide more sophisticated charging strategies, the use of active charging management systems is required. This allows considering concepts such as smart charging and vehicle to grid services [19,20], a solution of which is given by the offered services of the VPP operator. Possible enhancements are discussed in more detail in the following section.

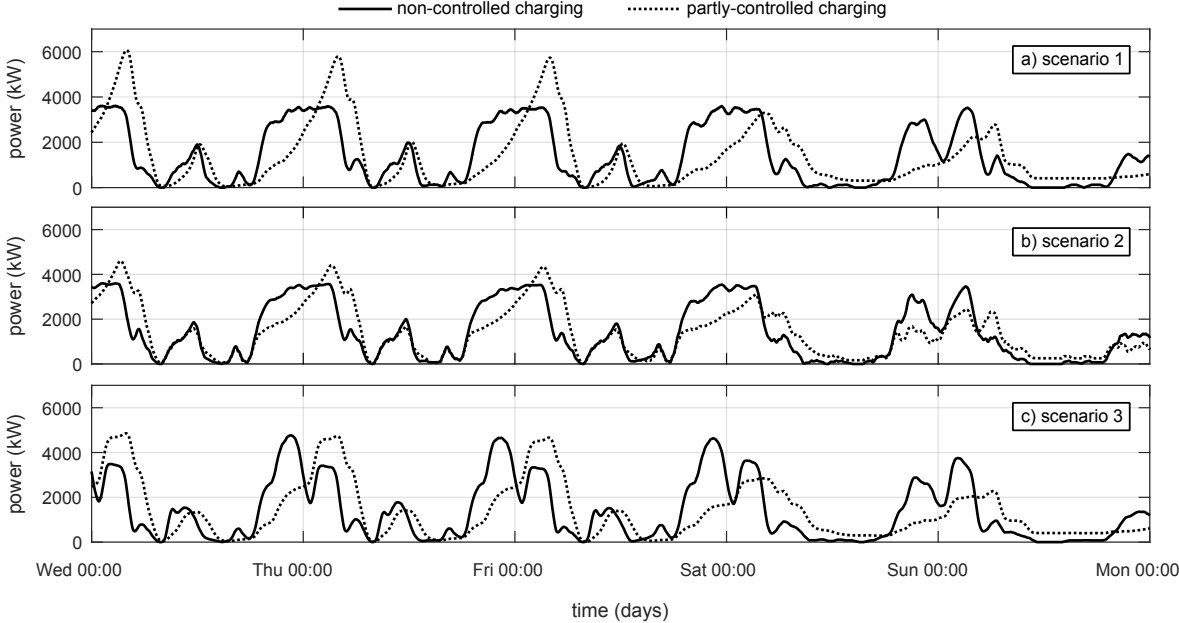

**Figure 8.** Charging load at the depot obtained by applying non-controlled and partly-controlled charging.

## 4. Optimized Energy Procurements in VPP Operations

The VPP operator integrates the electric bus fleet within the operational planning of its power plant portfolio as detailed in [21]. Let $H_{typ}$ be the set of unit types used in the power plant portfolio, consisting of the wind power plant (wind), photovoltaic power plant (pv), combined heat and power plant (chp), electric vehicles (ev), and industrial load units (ind). A multi-period optimization process is applied within the the energy management. This allows considering distinct forecast horizons $fh \in \{24\,h, 1\,h, 0.25\,h\}$ to determine mid-term and short-term bidding schedules. The classification of the forecast horizons is derived from the trading period and clearing sequence of joint market operations in day-ahead and intraday markets. The breakdown of generation and load schedules yields a more efficient use of the energy sources in market tradings [22]. Further, the VPP operator integrates shorter dispatch intervals to eliminate market framework barriers for the participation of renewable energy sources.

Using the information provided by the EVS/A and taking the depot characteristic discussed in Section 3, Table 2 gives a possible sample set used for the evaluation purposes of the proposed methodology. In this example, a total number of 193 buses is integrated into the power plant portfolio. The rated energy capacity $E_r^{fleet}$ of the bus batteries is 41.53 MWh. The forecasted daily energy demand $E_d^{fleet,1st}$ for the first-base charging assessments is 10.59 MWh.

**Table 2.** Composition of the power plant portfolio and electric bus fleet.

| Fleet Composition | | | | $E_d^{fleet,1st}$ | $E_r^{fleet}$ |
|---|---|---|---|---|---|
| **Total** | **SB** | **AB** | **DD** | **(MWh)** | |
| 193 | 53 | 103 | 37 | 10.59 | 41.53 |
| Installed Capacities of Generation, Load and Storage Units (MW) | | | | | |
| **Total** | **Wind** | **pv** | **chp** | **ev** | **ind** |
| 25 | 6 | 2.5 | 1 | 14.5 | 1 |

With regards to the integrated electric bus fleet, the VPP operator determines optimized charging schedules for each stage of the multi-period optimization process. This VPP service is provided to the EVS/A, which buys the electricity and responds to requests for the adjustment of charging schedules. First, the economic efficiency and feasibility of processing the optimized charging schedules are investigated. Then, the potentials for offering optimized redispatch measures are assessed as part of the extreme condition tests.

*4.1. Implementation Model and Mathematical Formulation*

The introduced unit models given in Table 1 are transferred into boundary and constraint conditions in the optimization model of the VPP operator. The boundary conditions are reflected by means of the provided ***TRIP*** and ***CON*** matrices of the EVS/A. The optimization problem combines the optimization variables given by the power dispatch $P_k^{typ}$ of all energy sources in the power plant portfolio, as well as the contracted market biddings $P_k^{em}$.

Therefore, the VPP operator applies (9), giving the cost-optimizing bidding strategy aiming to minimize the variable cost, while maximizing the relative gross profit. Hereby, $\omega_{vc}^{typ}$ indicates the variable operating cost for each unit type, while $\omega_{k,fh}^{em}$ defines the energy market price in day-ahead and intraday markets.

$$\min \sum - \left( \left( -P_{d,k}^{typ} \cdot \omega_{vc}^{typ} - P_{g,k}^{typ} \cdot \omega_{vc}^{typ} \right) - \left( P_{k,fh}^{em} \cdot \omega_{k,fh}^{em} \right) \right) \cdot \Delta t \quad \forall\, typ \in H_{typ} \tag{9}$$

The bidding strategy is subjected to the operating ranges $P_{g,min}^{typ} \leq P_{g,k}^{typ} \leq P_{g,max}^{typ}$ and $P_{d,min}^{typ} \leq P_{d,k}^{typ} \leq P_{d,max}^{typ}$ for the overall power generation and demand. In case of storage units including the electric bus fleet, the constraint formulation is given by:

$$y_k^{stor} \cdot P_{d,max}^{stor} \leq P_k^{stor} \leq (1 - y_k^{stor}) \cdot P_{g,max}^{stor}. \tag{10}$$

The binary variables $y_k^{typ}$ specify the operation mode of the distinct units. The available energy capacity values are derived from (11) as a function of the assigned rated energy capacity with regards to the vehicle models given in Table 1.

$$E_{k+1}^{mstor} = \begin{cases} E_k^{mstor} + P_{d,k}^{mstor} \cdot \eta \cdot \Delta t, & \text{charging mode} \\ E_k^{mstor} - P_{g,k}^{mstor} \cdot \frac{1}{\eta} \cdot \Delta t, & \text{discharging mode} \end{cases} \tag{11}$$

The dispatched power is bounded by the state of energy limits $SoE_{min}^{mstor} \leq SoE_{k+1}^{mstor} \leq SoE_{max}^{mstor}$, with $SoE_{k+1}^{mstor} = \frac{100\% \cdot E_{k+1}^{mstor}}{E_r^{mstor}}$. In each stage of the multi-period optimization process, the power balance $P_{g,k}^{VPP} = P_{d,k}^{VPP}$ of the power plant portfolio is calculated by applying (12) and (13).

$$P_{g,k}^{VPP} = P_{g,k}^{pv} + P_{g,k}^{wind} + P_{g,k}^{chp} + (1 - y_k^{fleet}) \cdot P_{g,k}^{fleet} + P_{IM,k}^{em} \tag{12}$$

$$P_{\mathrm{d},k}^{\mathrm{VPP}} = P_{\mathrm{d},k}^{\mathrm{ind}} + y_k^{\mathrm{fleet}} \cdot P_{\mathrm{d},k}^{\mathrm{fleet}} + P_{\mathrm{EX},k}^{em}. \tag{13}$$

The terms refer to the total power generation $P_{\mathrm{g},k}^{\mathrm{VPP}}$ and total power demand $P_{\mathrm{d},k}^{\mathrm{VPP}}$ of the power plant portfolio for each time step $k$. Market imports and market exports are indicated by $P_{\mathrm{IM},k}^{em}$ and $P_{\mathrm{EX},k}^{em}$, respectively.

### 4.2. Computational Study and Dispatch Results

The mixed-integer linear programming problem is solved by using a branch-and-cut method with simplex algorithm, offered by the MATLAB extension of the ILOG CPLEX optimization solver. In each stage, the bidding schedules are optimized while considering the unit type specific boundary and constraint conditions, including updated information and operational states formulated in Section 4.1. With a focus on the electric vehicle fleet, Figure 9 provides the obtained charging schedules for the first-base charging at depot. During day-ahead and intraday operation, the charging schedules are determined based on the forecasted power generation and demand of the installed renewable generation and load units within the power plant portfolio.

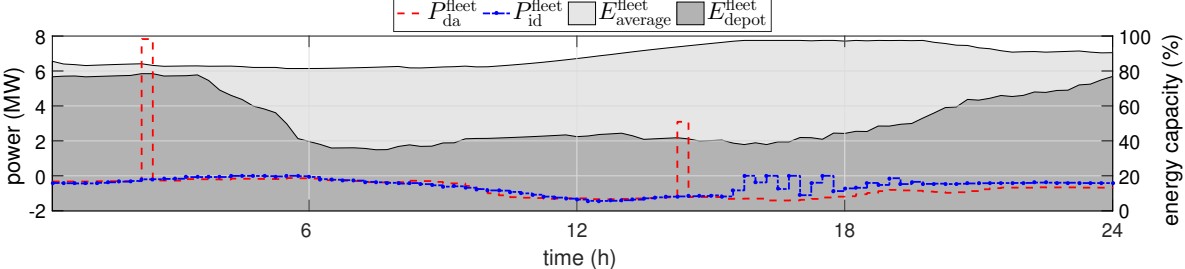

**Figure 9.** Day-ahead (da) and intraday (id) charging schedules and normalized energy capacity of the spatial and temporal available electric vehicle fleet.

For validation purposes, the average energy capacity of the electric bus fleet after the charging processes $E_{\mathrm{average}}^{\mathrm{fleet}}$, normalized on the rated energy capacity $E_{\mathrm{r}}^{\mathrm{fleet}}$, is given by the light gray area. As can be seen, the proposed methodology ensures the operability of the electric bus fleet by keeping the average energy capacity between 80% and 100%. The available energy capacity at the depot $E_{\mathrm{depot}}^{\mathrm{fleet}}$, normalized on the rated energy capacity $E_{\mathrm{r}}^{\mathrm{fleet}}$, is given by the dark gray area. At local peaks of the available energy capacity, specifically at 2:00 and 2:30, possible options for vehicle to grid operations are determined. However, during intraday operation, these services are not explicitly utilized due to updated information, e.g., requests for power system services. Besides, the charging schedule determined during intraday operation follows the day-ahead charging schedule, taking into account more precise forecasts of power generation and demand.

Addressing even more enhanced energy management and supply solutions by providing system services and redispatch measures, several positive CR+ and negative CR− control reserve requests of the system operator are investigated in the extreme condition test. The VPP operator reacts with optimized redispatch measures and calculates an alternative charging schedule. Figure 10 shows the charging schedules with and without considering the positive and negative control reserve requests. Here, the provision of 0.5 MWh (case 1), 1 MWh (case 2), and 2 MWh (case 3) through redispatch measures and hence adjusting the charging power is evaluated.

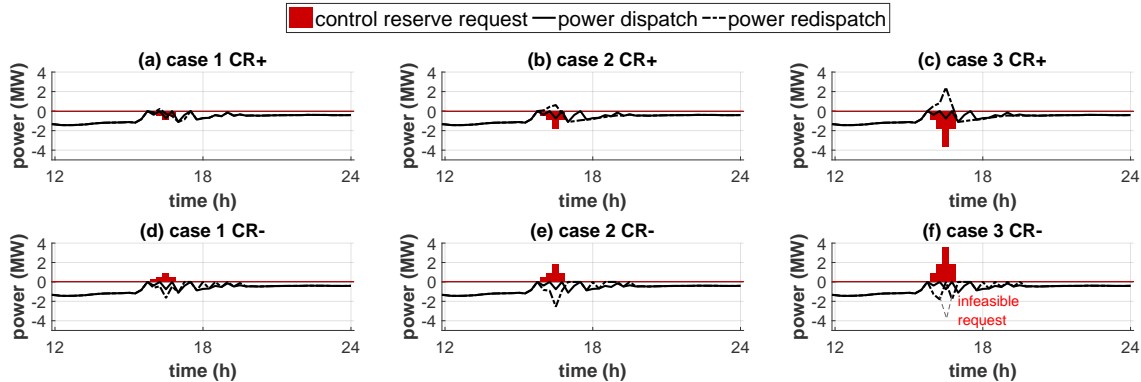

**Figure 10.** Response to positive and negative reserve power requests and performed redispatch measures.

The results show that every positive control reserve request can be fulfilled through charging power adjustments and vehicle to grid services. While this also applies to the first scenarios of negative control reserve requests, the peak request of 2 MWh cannot be fulfilled due to insufficient available negative reserve capacity. The infeasible solution is highlighted in Figure 10f. In this extreme condition test, the request of the system operator is denied by the VPP operator. In summary, the feasible solutions for the provision of system services and redispatch measures for an entire day are detailed by means of Figure 11, which shows the available reserve capacities at the depot. Giving insight into the simulation results obtained by testing the 2-MWh negative control reserve request, Figure 11d provides further details. The infeasible solution is caused due to the reduction of available negative reserve capacity, which is completely reduced to zero. The available positive control reserve requests remain the same for every case. This effect is due to the intended charging strategy that keeps the electric bus fleet at a high state of energy ranges, ensuring a high readiness for use of the electric bus fleet.

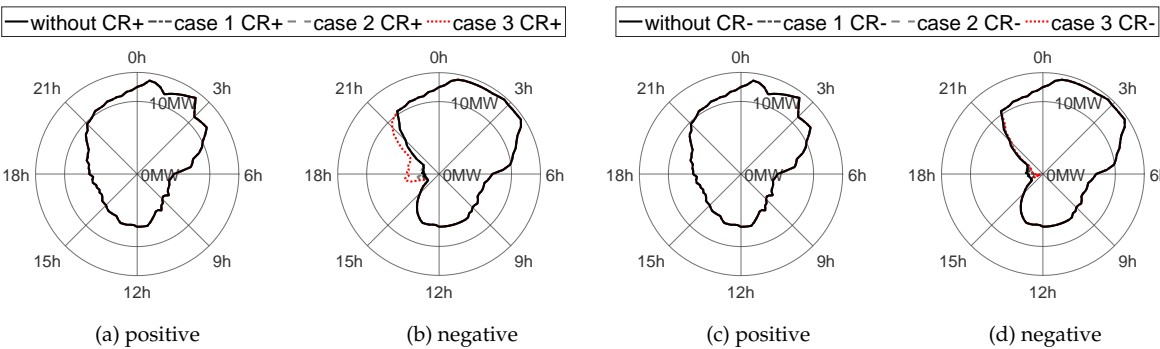

**Figure 11.** Available (**a**,**c**) positive and (**b**,**d**) negative reserve capacity of the electric bus fleet including the response to (**a**,**b**) positive and (**c**,**d**) negative reserve power requests.

The simulation results of the performed extreme condition tests prove the possibility for the provision of additional system services. This is achieved by optimally adjusting the charging schedules, while considering the boundary and constraint conditions, including updated information and the operational state for the operation of the electric bus fleet. Overall, the additional constraints given by the temporal availability and energy demand profiles of electric bus fleets are fully reflected in the optimization model. This allows achieving optimal charging solutions while fulfilling the contract position with the EVS/A. Further, the power provided by renewable energy sources can be optimally utilized for charging processes.

## 5. Conclusions

The paper addresses the specific challenges and opportunities arising with the presence of electric bus fleets in the operation scheme of the EVS/A and VPP operator in liberalized energy markets.

The impact on the overall energy supply is specified at the time of possible charging processes by assessing the operation procedures and depot characteristics. Enhanced charging strategies are developed by integrating the electric bus fleet in the energy management of a VPP operator. A comprehensive simulation framework is introduced. Through optimally-determining and adjusting the charging schedules in day-ahead and intraday operations, the energy demand for the operation of the electric bus fleet can be efficiently supplied within a multi-period optimization process. The iterative solution of the mixed integer linear programming problem allows a detailed representation of redispatch measures for the provision of power system services, while incorporating the constraints given by the electric bus fleet. The results show that the proposed methodology is capable of fully integrating electric bus fleets in the operation of a power plant portfolio, providing economic benefits for both the EVS/A and VPP operator.

**Author Contributions:** A.F.R. and E.L. conceived of the presented methodology. A.F.R. developed the framework introduced in Section 2, Framework Conditions and Operational Planning, and detailed the mathematical formulation. E.L. performed the evaluation discussed in Section 3, Operation Procedures and Depot Characteristics, and introduced the different charging scenarios. The numerical simulations conducted in Section 4, Optimized Energy Procurements in VPP Operations, were performed by A.F.R. K.S. and D.G. supervised the project and were involved in the review and editing process. All authors discussed the results and contributed to the final manuscript.

**Funding:** This research was funded by the Federal Ministry of Education and Research of the Federal Republic of Germany as part of the Research Campus Mobility2Grid (www.mobility2grid.de).

**Conflicts of Interest:** The authors declare no conflict of interest.

## Abbreviations

The following abbreviations are used in this manuscript:

| | |
|---|---|
| AB | articular bus |
| chp | combined heat and power plant |
| DD | double decker bus |
| dBus | diesel bus |
| eBus | electric bus |
| em | energy market |
| EV | electric vehicle |
| EVS/A | electric vehicle supplier/aggregator |
| ind | industrial load units |
| SB | standard bus |
| pv | photovoltaic power plant |
| VPP | virtual power plant |
| wind | wind power plant |

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
