# Peer review of "Implementation Schemes for Electric Bus Fleets at Depots with Optimized Energy Procurements in Virtual Power Plant Operations"

_wevj, doi:10.3390/wevj10010005_

Reviewer 1 Report

The work presented concerns a current theme of extreme importance. Electric energy systems are changing and the inclusion of EVs will be a future action that will bring about major changes. In the work is carried out a study of the integration of the electric bus fleets in the electrical systems with the market participation aided by a type of aggregator. I think it is a promising topic and currently has a lot of speculation and in the future will have much concern.

1 - The introduction refers to the problem and the author presents the solution, although I think the contributions could be further strengthened.

2 - A survey of the state of EV's integration should be carried out, at least in Europe. As well as what the state of adhesion of the companies to this type of vehicles.

3 - The second section presents many well-referenced theoretical concepts. The description should be simplified so that the reader has a better understanding.

4 - Section 3 presents a presentation of the case study although it has a different name. The figures are legibly readable.

5 - I think the results should appear in a single section. The presented results could be improved in order to demonstrate the real applicability of this methodology in a real environment.

6 - Conclusions should also be improved.

7 - A total of 7/20 references belong to the authors, this amount should be reduced.

Author Response

1 - The introduction refers to the problem and the author presents the solution, although I think the contributions could be further strengthened.

Thanks to the reviewer’s comment, we have rewritten the introduction to clarify this issue. The formulation regarding the contribution of the paper is updated as follows: Compared to the existing research as mentioned above and further aggregation and scheduling concepts in (Kang 2017, Liu 2015), the paper proposes a solution to integrate electric bus fleets in VPP operations. A methodology for the estimation of the energy demand is carried out by analyzing field-recorded data of diesel demands, determining the energy equivalence and forming bus type specific vehicle models. Furthermore, charging possibilities are identified that correspond to the operation conditions and services processes at a bus depot. As a result, optimal charging schedules are obtained while making use of these additional energy sources for energy market participation and provision of power system services. This is achieved thanks to novel VPP functions that exploit the potential of optimized redispatch solutions using the storage capacities of the electric bus fleets at a range of spatial and temporal scales.

2 - A survey of the state of EV's integration should be carried out, at least in Europe. As well as what the state of adhesion of the companies to this type of vehicles.

Additional descriptions regarding EV’s integration are reflected as follows: In multiple pilot projects different technologies for electric buses and charging systems are currently tested and demonstrated. Taking Europe for example, different bus types are utilized with conductive or inductive charging systems (Zeeus 2016, Kunith 2017). With focus on the operation of electric bus fleets, comprehensive analysis for the determination of network capacities and appropriate solutions for the power supply are required (Rogge 2015).  Here, several optimization techniques for smart charging strategies can be adapted to lower the overall energy cost and avoid grid congestion and peak loads caused by charging processes (Bessa 2012, Mets 2012, Zheng 2013, Shao 2016). Considering the charging load in enhanced energy management and supply solutions, it is necessary using predictive forecast methods to determine the energy demands.

3 - The second section presents many well-referenced theoretical concepts. The description should be simplified so that the reader has a better understanding.

A comprehensive summary of section 2 was conducted aiming to improve the readability. Major changes have been made by reducing the complexity, e.g. excluding the explanations related to the removed Fig. 2 that deals with the general description of the data handling and processing.

4 - Section 3 presents a presentation of the case study although it has a different name. The figures are legibly readable.

Section 3 identifies the fundamental characteristics of a depot, including services and processes impacting the charging process. The elaboration yields the input for the case study carried out in section 4. For clarification, further explanations are included in the respective sections.

5 - I think the results should appear in a single section. The presented results could be improved in order to demonstrate the real applicability of this methodology in a real environment.

The computational study provided in section 4 have been revised. Specifically, Fig 9. was modified providing the dispatch results for each stage of the multi-period optimization process. The real applicability is strengthened by comparing the day-ahead and intraday charging schedules and showing the charging status of the electric vehicle fleet during the operation.

6 - Conclusions should also be improved.

The conclusion section is improved addressing specific findings and highlighting the possible benefits for the EVS/A and VPP operator when considering electric bus fleets as an active element in power plant portfolio operation.

7 - A total of 7/20 references belong to the authors, this amount should be reduced

The author’s references are reduced to four. Additional references are added to cover the reviewer’s comments.

Reviewer 2 Report

The paper addresses the specific challenges and opportunities presented by the electrification of bus fleets. Comprehensive simulation frameworks are introduced addressing possible charging concepts and strategies for charging electric bus fleets. This paper has a certain contribution to the electric bus access to the power grid. Some comments are maded as following:

1.      In formula (5), the average diesel demand is obtained by the fleet's diesel demand, low calorific value and total mileage, and the unit kilometer diesel demand (l/km) is converted into unit kilometer energy demand (kWh/km). Is there a source? If there is any please mark it.

2.      Are the examples in Figure 5  the actual or the referenced bus operation? please explain or label accordingly.

3.      Is the comparison between the charging process in (b) of Figure 7 and the refueling process of the internal combustion bus reasonable? Can the current fast charging technology meet the requirements?

4.      If you can do a week of bus connectivity analysis in Figure 8, it is more convincing.

Author Response

1.      In formula (5), the average diesel demand is obtained by the fleet's diesel demand, low calorific value and total mileage, and the unit kilometer diesel demand (l/km) is converted into unit kilometer energy demand (kWh/km). Is there a source If there is any please mark it.

For the approximation of the required driving energy, field-recorded diesel demands of standard buses (SB), articulated buses (AB), and double decker buses (DD) are analyzed and the energy equivalents determined using the efficiency method. A linear regression model is initialized as a hypothesis concerning the relationship among the specific energy demand, use of auxiliary devices and the ambient temperature. Here, the assumption is made that the energy demand for driving is the same for diesel and electric buses as indicated by (6).

The lower calorific value of diesel states energy equivalent based on the thermodynamic quantity. Analogously to data used by the bus operator, the value is assumed with 9.94 kWh / l. This value is reasonable and in the same range as given in the literature, compare e.g. Demirbas (2017).

Demirbas (2017), Fuel Properties of Hydrogen, Liquefied Petroleum Gas (LPG), and Compressed Natural Gas (CNG) for Transportation, [Online] doi.org/10.1080/00908312.2002.11877434 [not included as reference]

2.      Are the examples in Figure 5 the actual or the referenced bus operation? please explain or label accordingly.

The explanation given in the text are revised and rewritten as follows: Real operating schedules over one year of a bus fleet are evaluated pointing out the difference in connectivity and energy demand for 1st-base and 2nd-base charging. Fig. 5a shows the results for a selected week of the entire electric bus fleet, while Fig. 5b and Fig. 5c gives further insight in the number of connected vehicles and corresponding energy demand at these locations.

3.      Is the comparison between the charging process in (b) of Figure 7 and the refueling process of the internal combustion bus reasonable? Can the current fast charging technology meet the requirements?

Scenario 2 is an extension of the previous scenario. It is assumed that fast charging during the service is possible. This is comparable with conventional refuel processes when considering vehicles with internal combustion engines. The following sentence has been added: Available and tested charging technologies enable conductive fast charging of up to 500 kW, demonstrated in European field sites (Zeeus 2016).

The daily service process lasts about 10-15 min (Lauber 2018). Thus, approximately 85 - 125 kWh can be charged during the service.

4.      If you can do a week of bus connectivity analysis in Figure 8, it is more convincing.

During the weekdays, the connectivity profile hardly changes because the operating schedules for Monday - Friday are identical. (…) Compared to the weekdays, the connectivity of the buses rises on the weekend, especially during the day, as the overall utilization of the electric bus fleet is lower. Therefore, just a selection of weekdays is chosen.